# Active Compounds in *Zingiber officinale* as Possible Redox Inhibitors of 5-Lipoxygenase Using an In Silico Approach

**DOI:** 10.3390/ijms23116093

**Published:** 2022-05-29

**Authors:** Jaqueline Stephanie Ley-Martínez, Jose Erick Ortega-Valencia, Oscar García-Barradas, Maribel Jiménez-Fernández, Esmeralda Uribe-Lam, Carlos Iván Vencedor-Meraz, Jacqueline Oliva-Ramírez

**Affiliations:** 1Laboratorio de Ingeniería de Superficies, Tecnológico de Monterrey, Escuela de Ingeniería y Ciencias, Av. Lago de Guadalupe Km. 3.5, Margarita Maza de Juárez, Ciudad López Mateos 52926, Mexico, Mexico; a01747661@itesm.mx; 2Instituto de Química Aplicada, Universidad Veracruzana, Av. Dr. Luis Castelazo s/n, Col. Industrial-Animas, Xalapa Enríquez 91190, Veracruz, Mexico; osgarcia@uv.mx; 3Centro de Investigación y Desarrollo en Alimentos, Universidad Veracruzana, Av. Dr. Luis Castelazo s/n, Col. Industrial-Animas, Xalapa Enríquez 91190, Veracruz, Mexico; maribjimenez@uv.mx; 4Tecnológico de Monterrey, Escuela de Ingeniería y Ciencias, México, Epigmenio González 500, Fraccionamiento San Pablo, Querétaro 76130, Querétaro, Mexico; euribelam@tec.mx; 5Research and Development Department, Genolife-Información de vida S.A.P.I de C.V., Blvd. Paseo Rio Sonora, Hermosillo 83270, Sonora, Mexico; vencedorcarlosipn@gmail.com

**Keywords:** 5-Lipooxygenase, 6-shogaol, 6-gingerol, inflammation, leukotrienes

## Abstract

5-Lipoxygenase (5-LOX) converts arachidonic acid to lipidic inflammatory mediators such as leukotrienes (LTs). In diseases such as asthma, LTs contribute to a physiopathology that could be reverted by blocking 5-LOX. Natural products with anti-inflammatory potential such as ginger have been used as nutraceuticals since ancient times. 6-Gingerol and 6-shogaol are the most abundant compounds in the ginger rhizome; they possess anti-inflammatory, antioxidant, and chemopreventive properties. In the present study, 6-gingerol and 6-shogaol structures were analyzed and compared with two commercial 5-LOX inhibitors (zileuton and atreleuton) and with other inhibitor candidates (3f, NDGA, CP 209, caffeic acid, and caffeic acid phenethyl ester (CAPE)). The pharmacokinetics and toxicological properties of 6-gingerol, 6-shogaol, and the other compounds were evaluated. Targeted molecular coupling was performed to identify the optimal catalytic pocket for 5-LOX inhibition. The results showed that 6-gingerol and 6-shogaol follow all of the recommended pharmacokinetic parameters. These compounds could be inhibitors of 5-LOX because they present specific interactions with the residues involved in molecular inhibition. The current study demonstrated the potential of 6-gingerol and 6-shogaol as anti-inflammatory agents that inhibit 5-LOX, as they present a high level of performance in the toxicological analysis and could be catabolized by the cytochrome p450 enzymatic complex; however, 6-gingerol was superior in safety compared to 6-shogaol.

## 1. Introduction

Inflammation is a natural immune response to adverse stimulation; during this process, molecular mediators are released to the extracellular space, and immune cells are recruited. The secreted molecules include cytokines and lipidic mediators that are derived from arachidonic acid (AA) [1,2]. The oxidation of polyunsaturated fatty acids such as AA, linoleic acid, docosahexaenoic acid, and eicosapentaenoic acid lead to fatty acid-controlled inflammation. The lipoxygenase (LOX), cyclooxygenase, and epoxygenase pathways mediate AA oxidation. One of the most critical lipooxygenases is 5-lipoxygenase (5-LOX) [1,2], a dioxygenase enzyme that contains a non-heme iron atom that is involved in the leukotriene (LT) pathway [2]. This enzyme catalyzes the formation of 5-(*S*)-hydroperoxy eicosatetraenoic acid (5-HPETE) from AA by incorporating molecular oxygen and performing the dehydration of 5-HPETE to leukotriene A4 [1]. In chronic inflammatory diseases such as rhinitis and rheumatoid arthritis, LTs are associated with specific G-protein-coupled receptors [3]. LTC4, LTD4, and LTE4 are associated with increased vascular permeability and bronchospasm. LTs participate in the physiopathology of asthma by promoting eosinophil infiltration into the lung, the induction of alveolar macrophage activation, and the stimulation of the increase of vascular permeability via histamine [1,3]. The deleterious effects that contribute to the pathology of asthma could be blocked by inhibiting 5-LOX. The active site of 5-LOX contains a targetable non-heme iron atom that is susceptible to redox inhibitors. Their mechanism of action involves the transition of ferric iron (Fe^3+^) to the inactive ferrous ion (Fe^2+^) [1,2]. Redox inhibitors such as zileuton atreleuton, BWb70c, NDGA, phenyl piperazine benzamides (3f), and coumaperines (CP 209) are 5-LOX inhibitors. Caffeic acid and CAPE compounds can eliminate radicals and alter the redox cycle of ferric ions. The only approved commercial inhibitors are zileuton and atreleuton [1,4]; however, these possess limitations. For example, zileuton presented a poor pharmacokinetic profile, a short half-life, and hepatotoxicity due to the presence of the thiophene fraction in its structure, which generates chemically reactive metabolites in the liver [1]. New 5-LOX inhibitors with better affinity or fewer side effects are needed [1,4,5].

Active compounds in the ginger rhizome (*Zingiber officinale*), 6-shogaol, and 6-gingerol, were recently reported to possess anti-inflammatory properties mediated by a reduction in tumor necrosis factor-α (Ι-κBα phosphorylation, NF-κB nuclear activation, and ΡΚC-α translocation), as well as antioxidant and chemopreventive properties [6,7]. Other than their immune-modulatory effects, there are no reports of their interactions with lipidic mediators [6]. In addition, previous studies have shown the inhibitory effect of 6-gingerol on the production of proinflammatory cytokines in murine peritoneal macrophages. Likewise, 6-shogaol has been shown to inhibit the gene expression of inducible nitric oxide synthase (iNOS) and cyclooxygenase (COX) induced by LPS in macrophages [7,8]. The purpose of the present study was to determine whether 6-gingerol and 6-shogaol would function as 5-LOX redox inhibitors to provide anti-inflammatory effects. We carried out an in silico analysis of the pharmacokinetic and toxicological properties compared with the other redox inhibitors (Figure 1). A molecular docking analysis was performed to evaluate the interactions of 6-gingerol and 6-shogaol with 5-LOX. Finally, we analyzed drug metabolism in silico via the interaction of CYP450 with the atomic sites of 6-gingerol and 6-shogaol [1,4].

## 2. Results

### 2.1. Prediction of Pharmacokinetic Properties

The pharmacokinetic properties analysis was obtained using the online calculation tools Molinspiration and Osiris Data Warrior software to evaluate 6-gingerol and 6-shogaol as inhibitors of 5-LOX compared with commercial inhibitors (zileuton and atreleuton) and other inhibitor candidates (Table 1). First, 6-gingerol and 6-shogaol must obey Lipinski’s rules that allow for the evaluation and monitoring of the drugs administered orally, in order to fulfill their pharmacological or biological functions. These five rules are as follows: (a) The molecular weight must be less than 500 g/mol; (b) the cLogP must be less than 5; (c) there must be five or fewer hydrogen bond donor sites; (d) there must be 10 or fewer hydrogen bond acceptor sites; and (e) there must be fewer than 10 rotatable bonds. Organic molecules that violate one of these rules may have bioavailability and pharmacological or biological action problems [8]. None of our molecules presented any rule violations (Table 1), suggesting that the molecules are suitable candidates for 5-LOX inhibition, and that they can be orally administered. Additionally, compared to the commercial inhibitors, the pharmacokinetic properties were similar to those of zileuton and atreleuton (Table 1).

The bioavailability of 6-gingerol and 6-shogaol was evaluated using TPSA analysis. This parameter is related to passive molecular transport through membranes, which permits the prediction of the drug transport properties and their bioavailabilities. The TPSA and the values of the rotatable bonds for gingerol were 66.76 Å^2^ and 10, respectively, while for 6-shogaol, they were 46.53 Å^2^ and 9, respectively (Table 1).

The percentage of absorption was calculated using TPSA [8,9,10] (Equation (1)). 6-Shogaol presented an absorption percentage of 92.95% according to the values obtained from Equation (1). The absorption percentage was higher than all of the other inhibitors (Table 1), except for candidate 3f, whose value was slightly higher. The behavior of the absorption percentage was inversely proportional to the value of the TPSA.

### 2.2. Drug Scoring and Toxicity Analysis

We used the Osiris Data Warrior computational tool and admetSAR to calculate the toxicity risk parameters such as mutagenicity, tumorigenicity, irritation, and the reproductive toxicities of 6-gingerol, 6-shogaol, and the other 5-LOX inhibitors (Table 2). The results were visualized using color codes, where green indicated low toxicity, yellow indicated intermediate toxicity, and red indicated high toxicity (Table 2). 6-Gingerol did not present any risk of toxicity, and 6-shogaol did not present a risk of tumorigenicity, irritant, or effect on reproduction; however, it presented a high possibility of mutagenicity because it possesses a double bond in its hydrocarbon chain.

To assess the overall potential of 6-gingerol and 6-shogaol as 5-LOX inhibitors, the overall drug score was calculated by combining the hydrophobicity, drug similarity, partition coefficient (cLogP), aqueous solubility, molecular weight, and toxicity parameters. 6-Gingerol and 6-shogaol presented pharmacological scores of 0.40 and 0.37, respectively. These results were superior to atreleuton and the candidates NDGA, caffeic acid, and CAPE.

### 2.3. Molecular Docking

Directed coupling was performed with the crystal structure of the 5-LOX/NDGA complex (pdb code: 6N2W) [2,11,12], the active site of which is composed of a tetrad of catalytic residues (His-367, His-372, His-550, and Leu-673) that coordinate with the iron atom, allowing for its catalysis [1,13]. Conformational analysis was performed to test molecular docking in 10 conformations of 6-gingerol and 6-shogaol. We looked for the best conformation with the lowest binding energy (ΔG, kcal/mol). 6-Gingerol exhibited a binding energy of −5.9 kcal/mol, while 6-shogaol exhibited a binding energy of −6.2 kcal/mol (Appendix A). Both compounds showed interactions with two histidines (His-367 with 4.91 Å for 6-gingerol and 3.27 Å for 6-shogaol; and His-372 with 4.43 Å for 6-gingerol and 3.54 Å for 6-shogaol) involved in the inhibition of 5-LOX. 6-Gingerol presented interactions with the residues Phe-359 (distance 4.03 Å), His-360 (distance 3.46 Å), Gln-363 (distance 1.98 Å), His-367 (distance 4.91 Å), Leu-368 (distance 3.48 Å), Ile-406 (distance 4.60 Å), Ala-410 (distance 3.51 Å), His-432 (distance 3.15 Å), Pro-569 (distance 3.53 Å), Arg-596 (distance 1.89 Å), Trp-599 (distance 4.40 Å), and His-600 (distance 2.53 Å) (Figure 2). 6-Shogaol displayed interactions with the residues Phe-359 (distance 5.14 Å), His-360 (distance 2.43 Å), Thr-364 (distance 2.54 Å), Leu-368 (distance 2.25 Å), Ala-410 (distance 4.67 Å), His-432 (distance 4.11 Å), Arg-596 (distance 1.99 Å), and His-600 (distance 2.87 Å) (Figure 3). 6-Gingerol and 6-shogaol were placed inside the catalytic pocket and 6-gingerol was stabilized by hydrogen bonding interactions with Gln-363, Arg-596, and His-600; while for 6-shogaol, interactions were made with Arg-596 and His-600. 6-Gingerol had π-π type interactions with the residues Phe-359 and Trp-599; and π-alkyl interactions with the residues His-360, His-367, Leu-368, His-372, Ile-406, Ala-420, and His-432. 6-Shogaol presented π-π type interactions with the Phe-359 residue; π-alkyl type interactions with the residues His-360, His-367, Leu-368, Ala-410, and His-432; and π-sigma interactions with the residue His-372. Hydrogen bonds and π-alkyl type interactions are the dominant interactions for stabilizing these compounds, and they can adequately interact with the active site of 5-LOX [4,5,6].

The interaction between 6-gingerol and 6-shogaol with 5-LOX showed high hydrophobicity on the surface, mainly in the alkyl zone of both molecules (Figure 4 and Figure 5). Figure 4 shows the hydrogen bond donor areas and the hydrogen bond acceptor areas where the hydroxyl group of the alkyl chain of 6-gingerol presents a hydrogen bond donor area with the Gln-363 residue, and the hydroxyl group of the aromatic ring presents a donor area with the residues Arg-596 and His-600 [5]. 6-Shogaol can function as a hydrogen donor, since it joins with Arg-596 and His-600 (Figure 5). 6-Gingerol and 6-shogaol are neutral; therefore, it can be inferred that they have zero charge (Figure 4 and Figure 5). In both cases, the values of the surface area that are accessible to the solvent for the surface of 6-gingerol and 6-shogaol are relatively high (Figure 4 and Figure 5).

### 2.4. In Silico Prediction of 6-Gingerol SOMs

To predict the atomic sites, and to calculate the probability of the cytochrome p450 enzymes (1A2, 2A6, 2B6, 2C8, 2C9, 2C19, 2D6, 2E1, and 3A4) undergoing metabolic modifications and metabolism mediated by human liver microsomes, we performed an in silico approach using the online tool Xenosite [14,15,16,17,18]. The results displayed the main metabolic sites within the chemical structures of 6-gingerol and 6-shogaol, represented using a color scale where red represents a greater probability of undergoing a metabolic transformation by any of the isoenzymes. At the same time, blue or white suggests no possibility of metabolism in that area (Figure 6 and Figure 7). According to Xenosite, the metabolic sites for 6-gingerol were the methyl of the methoxyl group, which has a high probability of being metabolized by all of the isoenzymes. CYP2A6 can metabolize the atomic site where the methylene and the methyl groups of the 6-gingerol alkyl chain are found (Figure 6). The atomic sites suggested for its metabolism in the structure of 6-shogaol are the methoxyl group that are attached to the aromatic ring, and the methylene and methyl groups that belong to the hydrocarbon chain of 6-shogaol (Figure 7).

## 3. Discussion

We found that the active compounds of *Zingiber officinale,* 6-gingerol, and 6-shogaol may display a redox inhibition of 5-LOX. Using an in silico evaluation of toxicity, the pharmacokinetic properties and their interactions with the catalytic site of 5-LOX showed that these compounds were suitable, and in some cases, displayed higher drug scores than the reported or approved commercial inhibitors [2,6,8]. Our analysis demonstrated that the two molecules followed the Lipinski rules, suggesting that the molecules can be orally administered and were similar to the pharmacokinetic properties of the compared inhibitors, as has previously been reported [2,4,8]. This finding was supported by the TPSA values obtained for both compounds; according to Veber’s rule, the permitted values for orally administrated drugs is TPSA ≤ 140 Å^2^, with a rotary bond value of only 10 [9].

In the toxicological evaluation, 6-gingerol was safer than 6-shogaol because it exhibited a higher degree of mutagenicity due to a double bond in its structure. This finding suggests that if 6-shogaol is going to be used as an inhibitor, the correct dose should be evaluated to avoid this toxicological risk, as previously reported [6,7].

Molecular docking showed that both 6-gingerol and 6-shogaol had a high affinity for the 5-LOX catalytic pocket as they had significant interactions with two (His-367 and His-372) out of the three histidines involved in the inhibition of 5-LOX [1,2,13]. 6-Shogaol demonstrated a lower binding energy because it showed a more significant interaction with the catalytic histidines for inhibition, and demonstrated higher degrees of interaction with the residues that were involved in this catalytic pocket [8,19].

Furthermore, the solvent-accessible surface area scores for 6-gingerol and 6-shogaol were relatively high (Figure 5). These values are related to the van der Waals forces between the ligands and 5-LOX, and the molecule buried in the protein (implying a receptor–ligand interaction). Other published works showing different targets for 6-gingerol or 6-shogaol attribute some of the properties obtained in our results to possible drugs [5,6,8,20]. Finally, in the metabolic evaluation, the compounds presented different atomic sites for catalysis as expected, these mainly being the methylene groups and the methoxyl group. Accordingly, these compounds could be catabolized safely by cytochrome p450 in the liver, without producing deleterious catabolites [6,14,21].

## 4. Materials and Methods

### 4.1. Calculation of Pharmacokinetic Parameters

The Molinspiration (https://www.molinspiration.com/ accessed on 20 April 2022) and Osiris Data Warrior (v 5.5.0) toolkits were used to verify the pharmacokinetic properties of the 6-gingerol, 6-shogaol, and 5-LOX inhibitors. The molecular descriptors were calculated, including the logarithm of the partition coefficient (cLogP), the number of hydrogen bond donors, number of hydrogen bond acceptors, molecular mass of the compounds, topological polar surface area (TPSA), number of rotatable bonds, and violations of Lipinski’s rule of five. Using the TPSA value, the percentage of absorption (% ABS) was calculated using the following equation [8,10]:(1)% ABS=109−(0.345×TPSA)

### 4.2. Calculation of Toxicity Potential

To calculate the toxicological properties of the 6-gingerol, 6-shogaol, and 5-LOX inhibitors, the Osiris Data Warrior program and the admetSAR (v2.0) online program were used. The attributes evaluated in each of the molecules were toxicity (mutagenicity, tumorigenicity, irritation, and reproductive effect), drug-likeness, and drug score [6,8].

### 4.3. In Silico Studies/Molecular Docking

#### Preparation and Catalytic Site Prediction of 5-LOX

The structure of the 5-LOX protein was obtained from the RCSB Protein Data Bank (PDB) under the PDB code 6N2W. This crystal structure comes with the NDGA ligand in order to perform a directed coupling at the catalytic site of the NDGA/6N2W complex [11,12].

The protein was prepared using the UCSF Chimera software (v1.16, San Francisco, California, USA), the water molecules were removed, and the missing hydrogen atoms were added. The active site features key residues such as His-367, His-372, His-550, and Leu-673, which are directly associated with 5-LOX inhibition [1].

### 4.4. Ligand Preparation

The two-dimensional structures of 6-gingerol, 6-shogaol, and the other redox molecules were elaborated using ChemDraw 8.0 (PerkinElmer Informatics, Waltham, MA, USA), and were imported into Avogadro (https://avogadro.cc accessed on 20 April 2022) to optimize the geometry using the force field function MMFF94. All compounds were saved as mol2 files for subsequent docking studies [19,22,23].

### 4.5. Docking Simulation

The standard procedure for molecular docking was carried out using a rigid protein from the RCSB Protein Data Bank with the code PDB 6N2W and a flexible ligand (6-gingerol and 6-shogaol) whose torsion angles were identified (for 10 independent runs per ligand). First, directed docking was performed using the ligand catalytic pocket (NDGA) that comes with the protein crystal structure, and molecular docking was performed in the UCSF Chimera program [11,12]. Polar hydrogens and partial Gasteiger charges were added, and a grid box was created using Autodock Vina tools at UCSF Chimera. The coupling results and analysis were visualized using Discovery Studio Visualizer (Biovia 2021). At the conclusion of the coupling, the best conformation for hydrogen bonding or π interactions were analyzed, including the binding energy of the free ligand (ΔG, kcal/mol) [4,22,23].

### 4.6. In Silico Prediction of Metabolism Sites (SOMs) Using the Xenosite Web Predictor

The Xenosite web predictor (https://swami.wustl.edu/xenosite/ accessed on 20 April 2022) allows for the prediction of potential SOMs in the chemical structures of xenobiotics and other small molecules [15,24]. It proposes the atomic sites of the molecules that were changed by nine primary isoenzymes of cytochrome p450 (CYP450): CYP1A2, CYP2B6, CYP2A6, CYP2C9, CYP2C8, CYP2D6, CYP2C19, CYP3A4, CYP2E1, and human liver microsomes. The software predicts the metabolism sites using a color scale (red to blue). Blue corresponds to zero probability, white represents random metabolism sites, and red is considered to be a metabolism site [14,15,24]. The chemical structures of 6-gingerol and 6-shogaol were uploaded to the website in SMILES format, and the prediction of the metabolic sites (SOM) was conducted [14,15].

## 5. Conclusions

6-Shogaol and 6-gingerol are suitable redox inhibitors of 5-LOX, due to the affinity of the catalytic pocket containing three histidine residues and the reduction of active iron (Fe^3+^) to its inactive form (Fe^2+^). The pharmacological values and toxicological analysis corresponded to a safe and orally administrable drug for 6-gingerol. Compared to the available 5-LOX inhibitors, 6-gingerol showed superior scores. By contrast, 6-shogaol exhibited a mutagenicity risk; therefore, more studies should be carried out, and the lethal doses must be adjusted in experimental assays to avoid deleterious effects. 6-Gingerol and 6-shogaol can be metabolized by CYP450 isoenzymes, primarily in the methoxyl groups and the methylene and methyl groups of the hydrocarbon chain.

## Figures and Tables

**Figure 1 ijms-23-06093-f001:**
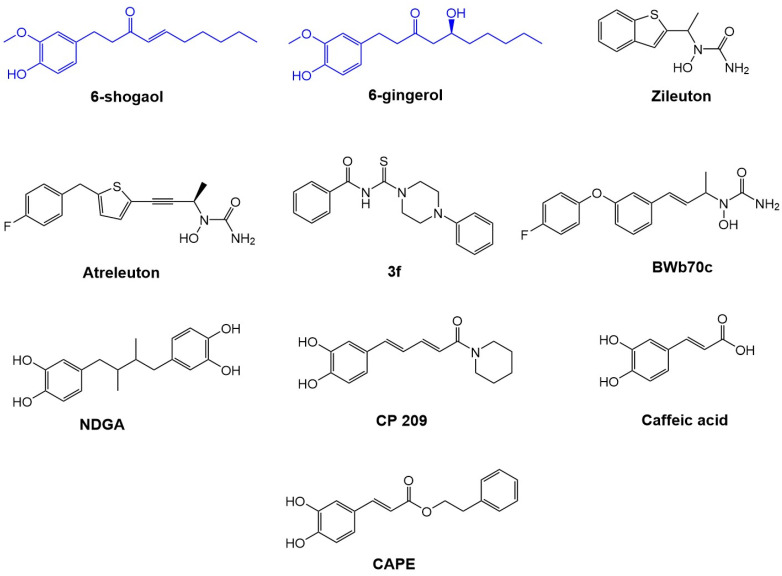
Chemical structures of main redox inhibitors of 5-LOX. Chemical structures of 6-gingerol, 6-shogaol, commercial redox inhibitors (zileuton and atreleuton) of 5-LOX, and candidate inhibitors of 5-LOX.

**Figure 2 ijms-23-06093-f002:**
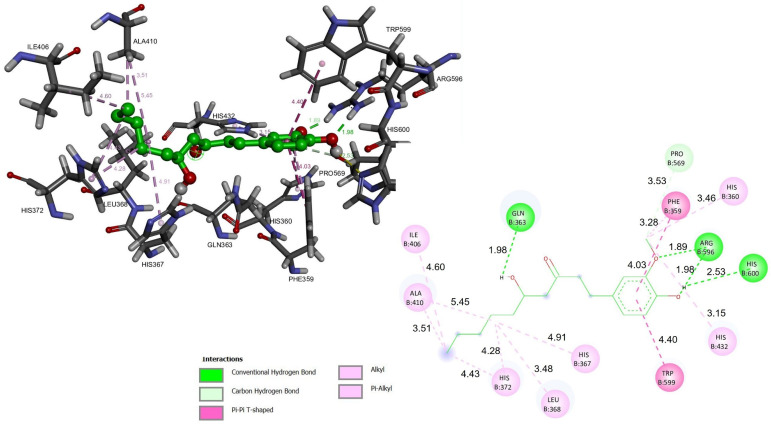
Molecular docking of 6-gingerol with 5-LOX (PDB:6N2W). The main residues involved in the protein–ligand interaction (**left**) and the main interactions involved in the protein–ligand binding (**right**) of 6-gingerol; distances are shown as Å over the lines.

**Figure 3 ijms-23-06093-f003:**
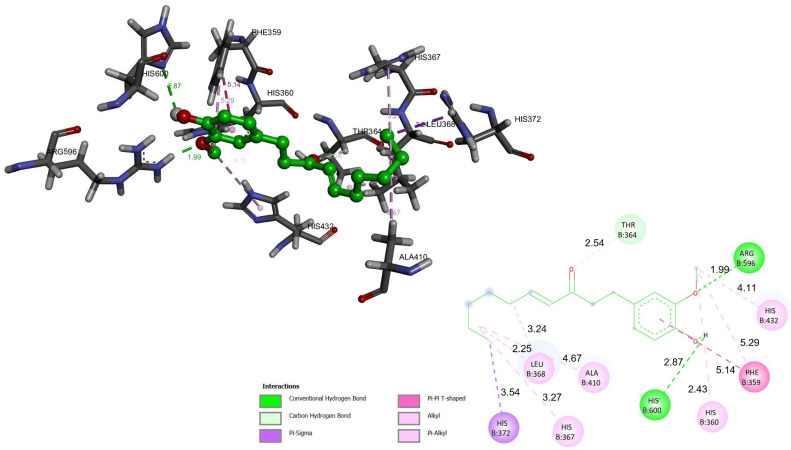
Molecular docking of 6-shogaol with 5-LOX (PDB:6N2W). The main residues involved in protein–ligand interactions (**left**) and the main interactions involved in protein–ligand binding (**right**) of 6-shogaol, distances are showed as Å over the lines.

**Figure 4 ijms-23-06093-f004:**
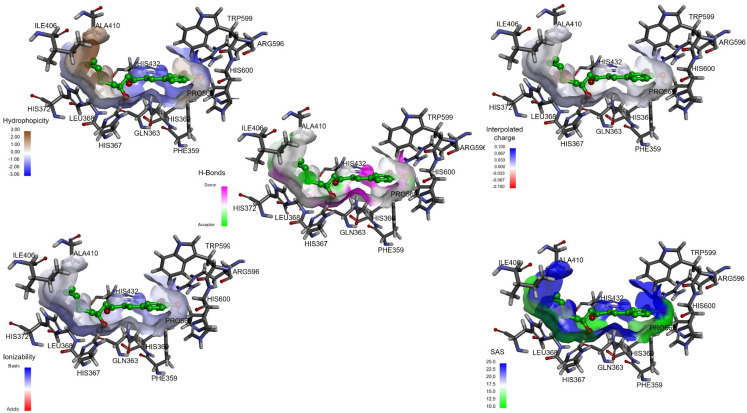
Interactions between the surface of 6−shogaol and 5−LOX (PDB:6N2W). Physicochemical complex of 6−shogaol and 5−LOX evaluating the hydrophobicity, hydrogen bonds, active lateral charge, ionization, and solvent-accessible surface (SASA).

**Figure 5 ijms-23-06093-f005:**
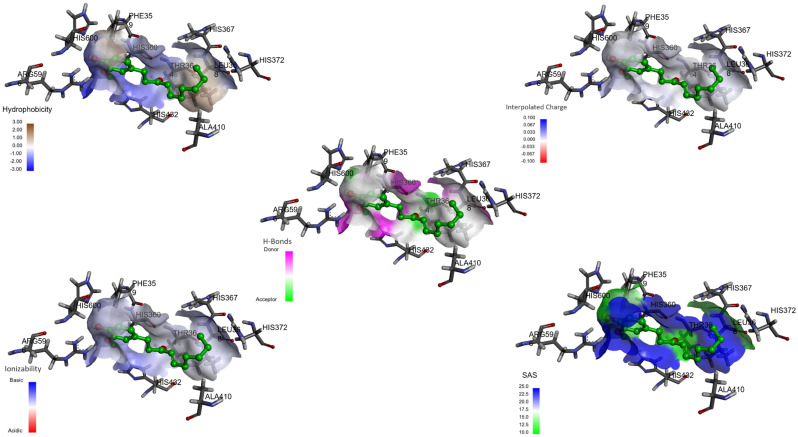
Interactions between the surface of 6-shogaol and 5−LOX (PDB:6N2W). Physicochemical complex of 6−shogaol and 5−LOX evaluating the hydrophobicity, hydrogen bonds, active lateral charge, ionization, and solvent-accessible surface (SASA).

**Figure 6 ijms-23-06093-f006:**
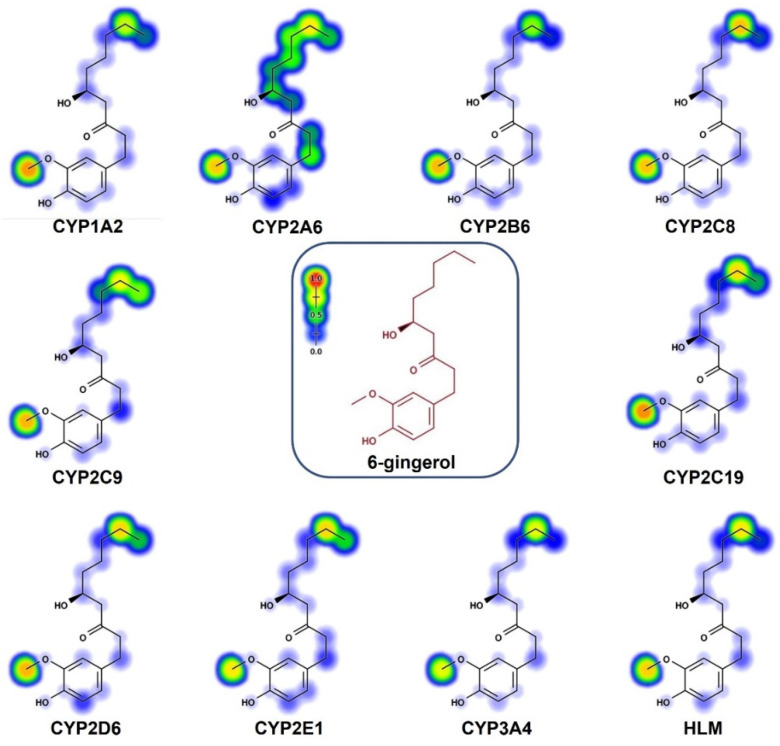
In silico evaluation of the metabolism of 6-gingerol. Metabolic sites (SOM) of 6-gingerol mediated by CYP450 isoenzymes (CYP1A2, CYP2A6, CYP2B6, CYP2C8, CYP2C9, CYP2C19, CYP2D6, CYP2E1, CYP3A4, and HLM).

**Figure 7 ijms-23-06093-f007:**
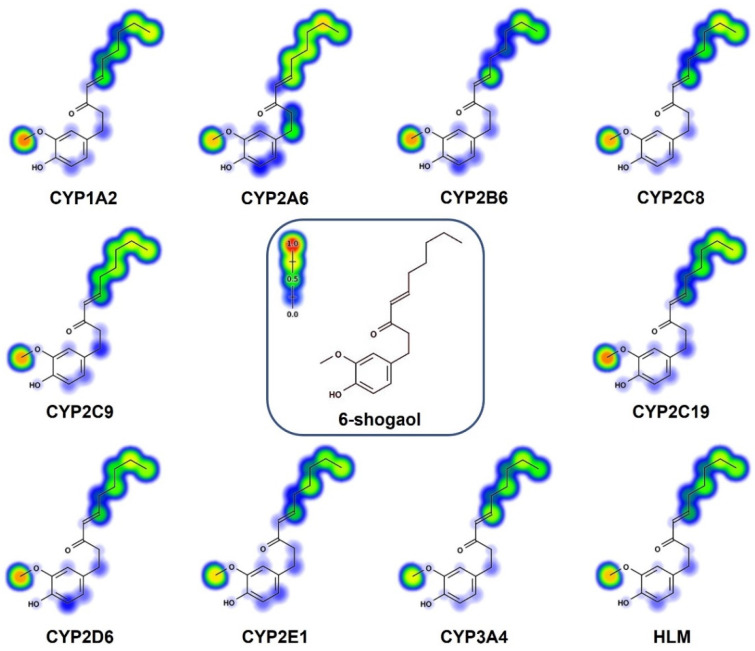
In silico evaluation of the metabolism of 6-shogaol. Metabolic sites (SOM) of 6-shogaol mediated by CYP450 isoenzymes (CYP1A2, CYP2A6, CYP2B6, CYP2C8, CYP2C9, CYP2C19, CYP2D6, CYP2E1, CYP3A4, and HLM).

**Table 1 ijms-23-06093-t001:** Pharmacokinetic properties of 6-shogaol and 6-gingerol compared with redox inhibitors of 5-LOX obtained from Osiris Data Warrior and Molinspiration software.

Compound	%ABS ^a^	TPSA (Å^2^) ^b^	MW ^c^	cLogP ^d^	HBD ^e^	HBA ^f^	*n*-ROTB ^g^	Violation of Lipinski’s Rule
Rule	-	-	<500	≤5	≤5	≤10	≤10	≤1
6-Shogaol	92.95	46.53	276.37	4.33	1	3	9	0
6-Gingerol	85.97	66.76	294.39	3.56	2	4	10	0
Zileuton *	76.30	94.80	236.29	1.23	2	4	2	0
Atreleuton *	86.04	66.56	318.10	2.94	2	4	3	0
3f	96.73	35.57	325.43	3.12	1	4	4	0
BWb70c	82.85	75.79	316.33	2.63	2	5	5	0
NDGA	81.09	80.91	302.37	3.82	4	4	5	0
CP 209	88.03	60.77	273.33	2.79	2	4	3	0
Caffeic acid	82.18	77.75	180.16	0.78	3	4	2	0
CAPE	85.97	66.76	284.31	3.05	2	4	6	0

^a^ Percentage of absorption (%ABS); ^b^ topological polar surface area (TPSA); ^c^ molecular weight (MW); ^d^ logarithm of partition coefficient between *n*-octanol and water (cLogP); ^e^ number of hydrogen bond donors (HBD); ^f^ number of hydrogen bond acceptors (HBA); ^g^ number of rotatable bonds (*n*-ROTB). * Commercial redox inhibitors of 5-LOX.

**Table 2 ijms-23-06093-t002:** Evaluation of drug-likeness, drug score, and toxicity risks of 6-shogaol and 6-gingerol compared to 5-LOX redox inhibitors using Osiris Data warrior software.

Compound	Mutagenic	Tumorigenic	Irritant	Reproductive Effect	Solubility	Drug-Likeness	Drug Score
6-Shogaol	Red	Green	Green	Green	−3.42	−15.81	0.37
6-Gingerol	Green	Green	Green	Green	−3.25	−9.06	0.40
Zileuton *	Green	Green	Green	Green	−3.24	1.84	0.86
Atreleuton *	Green	Green	Green	Green	−5.24	−2.98	0.35
3f	Green	Green	Green	Red	−3.22	4.95	0.49
BWb70c	Green	Green	Green	Green	−5.32	−0.24	0.51
NDGA	Green	Red	Green	Green	−2.93	−2.42	0.34
CP 209	Green	Green	Green	Green	−2.31	0.53	0.81
Caffeic acid	Red	Red	Green	Red	−1.41	0.17	0.19
CAPE	Green	Green	Red	Green	−2.97	−3.24	0.29

Green color shows low toxicity tendency, and red color shows high tendency of toxicity. * Commercial redox inhibitors of 5-LOX.

## Data Availability

Not applicable.

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
