# Peer review of "Active Compounds in Zingiber officinale as Possible Redox Inhibitors of 5-Lipoxygenase Using an In Silico Approach"

_ijms, 2022, doi:10.3390/ijms23116093_

Round 1
Reviewer 1 Report
I reviewed the manuscript “Active compounds in Zingiber officinale as possible redox inhibitorsof 5-lipoxygenase using an in-silico approach” by Ley-Martínez at all. The manuscript represents an in-silico approach to investigate a potential of natural compounds such as 6-gingerol and 6-shogaol to act as inhibitors of 5-lipoxygense (5-LOX). The work could be of interest to medicinal chemists, however, I could not recommend the manuscript for publication. Here are some problematic parts:
- Abstract, line 24-25: MK-591 and MK-886 are not direct 5-LOX inhibitors. They are inhibitors of the 5-lipoxygenase-activating protein (FLAP) which is a helper protein necessary for the efficient enzyme utilization of the endogenous substrate. Those compounds could not be used in current study since they do not target the active site of 5-LOX. The same applies to the Page 2 line 57: MK-591 and MK-886 should be removed. All mentions of MK-591 and MK-886 should be removed across the entire body of manuscript as well (several more instances).
- Figure of chemical structures of inhibitors should be in the main body of the manuscript for clarity, not in the Supplemental Materials section.
- The crystal structure of 5-LOX/NDGA complex is published (pdb code: 6N2W). The specific interactions between the inhibitor and 5-LOX are known. However, the authors of present study failed to discuss them in their manuscript.
- The major problem in the manuscript. Structure of 5-LOX is known in two different conformations: closed (pdb code: 3O8Y) and open one (pdb code 6N2W; complex with NDGA). They differ in location of loops surrounding the active site. Only the open conformation (6N2W) represents the active site that accommodates an inhibitor. Hence, 6N2W should be used for any modeling of active site interactions with potential inhibitors, NOT the 3O8Y, that was used in the study.
- Page 3, line 117: there is only one catalytic pocket in the 5-LOX molecule, which contains a non-heme iron. The other possible sites could be called “binding” sites etc. but not the “catalytic” ones. Description of the pockets is practically non-existing in the manuscript. Figure 1-3 are not possible to read, they bear almost no information. Figures are too small. The reader would want to see hydrogen bond distances (numbers) for interactions between inhibitors molecules and amino acids of the protein, as well as hydrophobic interactions (what cut-off is used in the manuscript to determine such interactions?). Such information absents in the manuscript.
- Page 5, line 199 : From the spatial position of the catalytic His367, His372 and the residue His-600 it is hard to believe that 6-shagaol can make H-bonding interactions with all of them. No intermolecular interactions are given (distances)
Author Response
Thank you for all your commentaries, please see the attachment, in which we answered and included more analysis.
Dear Dr. Maurizio Battino
Editor-in-Chief
We thank the reviewers for their time and effort in reviewing and evaluating our manuscript. We greatly appreciate your comments to significantly improve our manuscript. Line-by-line answers are provided below to the comments, and we incorporated the suggestions in the current version of our manuscript, we strongly believe that with all the improves made you will find it suitable for approval.
Editor and Reviewer comments:
Reviewer #1: Manuscript ID: ijms-1668787
- Abstract, line 24-25: MK-591 and MK-886 are not direct 5-LOX inhibitors. They are inhibitors of the 5-lipoxygenase-activating protein (FLAP) which is a helper protein necessary for the efficient enzyme utilization of the endogenous substrate. Those compounds could not be used in current study since they do not target the active site of 5-LOX. The same applies to the Page 2 line 57: MK-591 and MK-886 should be removed. All mentions of MK-591 and MK-886 should be removed across the entire body of manuscript as well (several more instances).
ANSWER: Thank you for your comments. We carefully consider your suggestion. Indeed, the compounds MK-591 and MK-886 are direct inhibitors of the 5-lipooxygenase activating protein (FLAP), since they are not direct inhibitors of 5-LOX, we have decided to remove them from the entire manuscript as previously requested.
Summary of the changes made in the manuscript: The compounds MK-591 and MK-886 were removed from the manuscript, specifically in lines 24, 25, 57 and 120, in the same way they have been removed from tables 1 and 2.
- Figure of chemical structures of inhibitor should be in the main body of the manuscript for clarity, not in the Supplemental Materials section.
ANSWER: Thank you very much for the suggestion to improve the structure of the manuscript. Therefore, the figure of the chemical structures of the inhibitors was added in the body of the manuscript instead of being in the supplementary materials section.
Summary of the changes made in the manuscript: We included the figure of the chemical structures of the inhibitors in the manuscript, the number of the figure was added as Figure 1 on line 76 page 2. In addition, the number of figures with respect to this movement, therefore, the number of figures in lines 129, 136, 139, 152, 160, 161, 162, 167, 173, 174, 178, 179, 181, 190, 193, 196 and 219 was modified
- The crystal structure of 5-LOX/NDGA complex is published (pdb code: 6N2W). The specific interactions between the inhibitor and 5-LOX are known. However, the authors of present study failed to discuss them their manuscript.
ANSWER: Thank you very much for your punctual comment, according to what you comment, we have decided to incorporate the specific interactions already known in 5-LOX/NDGA as well as the catalytic pocket for this crystalline structure to improve our molecular docking analysis; by verifying that the predicted pocket in the 3O8Y protein presents the specific interactions for its inhibition.
Summary of changes made to the manuscript: We include three lines (123-125, 133-134) where we comment that we use as a reference the known binding site in the 5-LOX/NDGA complex (pdb code: 6N2W).
- The major problem in the manuscript. Structure of 5-LOX is known in two different conformations: closed (pdb: 3O8Y) and open one (pdb code: 6N2W; complex with NDGA). They differ in location of loops surrounding the active site. Only the open conformation (6N2W) represents the active site that accommodates an inhibitor. Hence, 6N2W should be used for any modeling of active site interactions with potential inhibitors, NOT the 3O8Y, that was used in the study.
ANSWER: Thank you, we appreciate your comment for this reason, we have decided to carry out a directed docking at the catalytic pocket where the NDGA is found in the 5-LOX crystal structure (pdb code: 6N2W) with the aim of see the specific interactions and have them as a reference and comparison with the docking of our study with the crystal structure that does not have an incorporated ligand (pdb code: 3O8Y).
Summary of changes made to the manuscript: To enrich this analysis, the results of the directed docking with the 5-LOX/NDGA complex are incorporated as Supplementary Figure 1, in the same way the main interactions involved in this docking are incorporated, comparing it with the docking of our study, this information was incorporated in the manuscript on lines 138, 139, 140, 141, 147, 148, 149, 150, 151 and 152. In addition, the results of the directed docking of 6-gingerol and 6-shogaol with the crystal structure of the 5-LOX/NDGA complex (pdb code: 6N2W) are annexed to this document. These results were annexed to the manuscript as Supplementary Figure 1.
- Page 3, line 117: there is only one catalytic pocket in the 5-LOX molecule, which contains a non -heme iron. The other possible sites could be called “binding” sites etc. but not the “catalytic” ones. Description of the pockets is practically non-existing in the manuscript. Figures are too small. The reader would want to see hydrogen bond distances (numbers) for interactions between inhibitors molecules and amino acids of the protein, as well as hydrophobic interactions (what cut-off is used in the manuscript to determine such interactions?). Such information absents in the manuscript.
ANSWER: We kindly thank the reviewer, according to this comment, the change of “catalytic pockets” to “binding pockets” was made. 6-gingerol and 6-shogaol docking distances and interaction diagrams were plotted to visualize hydrogen bonding and hydrophobic interactions.
Summary of changes made to the manuscript: We have made the change of catalytic pockets as binding pockets in lines 122, 124, 126, 126, 129 and Figure 4 was modified by accommodating the molecular docking so that the interactions were also better visualized. that the distances in each of the molecular couplings were added
- Page 5, line 199: From the spatial position of the catalytic His367, His372 and the residue His600 it is hard to believe that 6-shogaol can make H-bonding interactions with all of them. No intermolecular interactions are given (distances).
ANSWER: We appreciate your comment, since the interaction that the His-600 residue presents with the 6-shogaol is an interaction of the carbon-hydrogen bond type and not an interaction of the H bond type.
Summary of changes made to the manuscript: We add three lines (16, 170 and 171) where we explain that there is a carbon-hydrogen bond type interaction between the His-600 residue and the compounds to be evaluated (6-gingerol and 6-shogaol).
List of Changes ijms-1668787
Reviewer 1
- The compounds MK-591 and MK-886 were removed from the manuscript, in the same way they have been removed from tables 1 and 2.
- Include the figure of the chemical structures of the inhibitors in the manuscript.
- Include as a reference the known binding site in the 5-LOX/NDGA complex (pdb code:6N2W).
- Include as a Supplementary Figure the results of the directed coupling between the compounds (6-gingerol and 6-shogaol) and the crystal structure of the 5-LOX/NDGA complex to use as a reference in our molecular couplings.Include a line in the manuscript with information about the considerations for selecting typhimurium concentration in this work.
- Change to the catalytic pockets as binding pockets
- Change the H bond interaction to carbon hydrogen bond interaction with the His-600 residue
Reviewer 2 Report
The manuscript entitled “Active compounds in Zingiber officinale as possible redox inhibitors of 5-lipoxygenase using an in-silico approach” by Ley-Martínez et al. demonstrates that 6-gingerol and 6-
shogaol, from Zingiber officinale, may display redox inhibition of 5-LOX. The study is well organized and
written; methods are well described (except for equation 1, which should be formatting according to the
text), the experiments seem to be carefully performed, and results and discussion are well-argued. Based
on this evidence, the manuscript can be considered suitably innovative and significant for publication. I would suggest to underline more the importance of this study in the state of the art of the contest.
Author Response
We thank the reviewers for their time and effort in reviewing and evaluating our manuscript. We greatly appreciate your comments to significantly improve our manuscript. Line-by-line answers are provided below to the comments, and we incorporated the suggestions in the current version of our manuscript, we strongly believe that with all the improves made you will find it suitable for approval.
- The study is well organized and written; methods are well described (except for equation 1, which should be formatting according to the text), the experiments seem to be carefully performed, and results and discussion are well-argued. Based on this evidence, the manuscript can be considered suitably innovative and significant for publication. I would suggest to underline more the importance of this study in the state of the art of contest.
ANSWER: We greatly appreciate your comments and observations, we modified the reference of equation 1 to the font used in the manuscript. As well we added more information about the importance of ginger's active compounds and recent studies about the anti-inflammatory properties that have been conducted.
Summary of changes made to the manuscript: The following data from recent studies was added on lines 69-72 about the importance of the active principles of ginger.
“In addition, recent studies have shown the inhibitory effect of 6-gingerol on the production of proinflammatory cytokines in murine peritoneal macrophages. Likewise, 6-shogaol has been shown to inhibit the gene expression of inducible nitric oxide synthase (iNOS) and cyclooxygenase (COX) induced by LPS in macrophages.”
It was made the modification of equation 1 in the text on line 233.
Round 2
Reviewer 1 Report
I reviewed the revised version of the manuscript “Active compounds in Zingiber officinale as possible redox inhibitors of 5-lipoxygenase using an in-silico approach” by Ley-Martínez at all. I still could not recommend the manuscript for the publication in the current form.
- The two published 5-LOX structures are very different in the vicinity of the active site. The active site of 3O8Y is very disordered. Amino acid residues 170-210, 294-303 and 416-429 are absent in the pdb-file. On the contrary, the 6N2W complex of 5-LOX with NDGA has those parts ordered and those residues are in the pdb-file. Hence, protein coordinates only from the 6N2W structure should be used for the docking analysis. There is no sense to use coordinates of the 3O8Y because of different conformation and extensive disorder in the active site. So, any data from the 3O8Y structure for docking analysis should be removed from the manuscript and docking must be performed using protein coordinates from the 6N2W structure.
- The authors use the dimer of 5-LOX in all their figures (Fig.2, 3,4). They should use only a monomer of 5-LOX for illustrations. The active site is the same in both monomers and using just a monomer would make figures clearer. Also, the authors did not improve most of their figures. Fig.2 does not carry any explanation of the “catalytic” pockets. Page 3, line 139: they refer to His-367, His-372 and His-600 on the Fig 2b. How could the reader even see those residues in the figure? I could not at all. Fig.4 is still impossible to read: amino acids labels are on top of the sticks representing amino acids, most of the intermolecular distances are not readable. The same addresses to the Fig.5. and Suppl. Fig.1.
- Page 4, line 151. The authors describe the catalytic residues of 5-LOX as the triad of His-367, His-372 and His-600, which is absolutely wrong. The catalytic residues of 5-LOX are well established and consists of the tetrad: His-367, His-372, His-550 and Leu-673. The His-600 does NOT coordinates to the iron atom!
Author Response
We kindly appreciate all your commentaries, please see the attachment.
- The two published 5-LOX structures are very different in the vicinity of the active site. The active site of 3O8Y is very disordered. Amino acid residues 170-210, 294-303 and 416- 429 are absent in the pdb-file. On the contrary, the 6N2W complex of 5-LOX with NDGA has those parts ordered and those residues are in the pdb-file. Hence, protein coordinates only from the 6N2W structure should be used for the docking analysis. There is no sense to use coordinates of the 3O8Y because of different conformation and extensive disorder in the active site. So, any data from the 3O8Y structure for docking analysis should be removed from the manuscript and docking must be performed using protein coordinates from the 6N2W structure.
Response: Thanks for your comments. We have considered your suggestion and we have decided to change the molecular coupling by substituting the crystal structure with the code PDB: 3O8Y for the crystal structure with the code PDB: 6N2W which, as mentioned above, has a ligand (NDGA) which allows us to specifically identify the coordinates of the catalytic site and perform the targeted coupling on that active site.
Summary of the changes made in the manuscript: The identification process of the catalytic pockets using the DogSiteScorer program was removed from the manuscript, in the same way everything related to the PDB:3O8Y crystal structure was removed from the manuscript. Specifically in lines 122, 142, 148 and 154 were eliminated and figures 2, 3, 4 and 5 were modified.
- The authors use the dimer of 5-LOX in all their figures (Fig.2, 3,4). They should use only a monomer of 5-LOX for illustrations. The active site is the same in both monomers and using just a monomer would make figures clearer. Also, the authors did not improve most of their figures. Fig.2 does not carry any explanation of the “catalytic” pockets. Page 3, line 139: they refer to His-367, His-372 and His-600 on the Fig 2b. How could the reader even see those residues in the figure? I could not at all. Fig.4 is still impossible to read: amino acids labels are on top of the sticks representing amino acids, most of the intermolecular distances are not readable. The same addresses to the Fig.5. and Suppl. Fig.1.
Response: We appreciate your comments. The pertinent changes were made in the figures, with the aim of better observing the interactions, the residues and the distances involved in each of the molecular couplings. Similarly, supplementary figure 1 was eliminated from supplementary figures and included in the manuscript, with improves in the visualization and include the distances values.
Summary of the changes made in the manuscript: We included Supplementary Figure 1 within the manuscript and improved its visualization. The images of figures 2, 3, 4 and 5 were enlarged so that the residues involved in the coupling could be observed, as well as the types of interactions and their respective distances added from lines 129 to 138.
- Page 4, line 151. The authors describe the catalytic residues of 5-LOX as the triad of His-367, His-372 and His-600, which is absolutely wrong. The catalytic residues of 5-LOX are well established and consists of the tetrad: His-367, His-372, His-550 and Leu-673. The His-600 does NOT coordinates to the iron atom!
Response: Thanks for your comments. The references were reviewed in detail and indeed we have observed that the catalytic site in the crystal structure with code PDB: 6N2W has a tetrad of residues involved in its inhibition (His-367, His-372, His-550 and Leu-673) therefore, we have decided to modify our information on the residues involved in the inhibition, relying on our references, in addition to carrying out the relevant discussion on the molecular coupling with the new crystal structure.
The selected references used were: Gilbert NC, Gerstmeier J, Schexnaydre EE, et al. Structural and mechanistic insights into 5-lipoxygenase inhibition by natural products. Nat Chem Biol. 2020;16(7):783-790. doi:10.1038/s41589-020-0544-7
Chiasson AI, Robichaud S, Ndongou Moutombi FJ, et al. New zileuton-hydroxycinnamic acid hybrids: Synthesis and structure-activity relationship towards 5-lipoxygenase inhibition. Molecules. 2020;25(20):1-18. doi:10.3390/molecules25204686
Sinha S, Doble M, Manju SL. 5-Lipoxygenase as a drug target: A review on trends in inhibitors structural design, SAR and mechanism based approach. Bioorganic Med Chem. 2019;27(17):3745-3759. doi:10.1016/j.bmc.2019.06.040
Summary of changes made to the manuscript: Lines 124, 129-130, 188-190, and 225 to 227 were modified
Round 3
Reviewer 1 Report
The manuscript is greatly improved and could be recommended for publication after minor corrections:
- Line 122-123: ".... the crystal structure of the 5-LOX/NDGA complex which is composed of tetrad ...". It is not the crystal structure, which is composed of tetrad of catalytic residues. It is the active site, which is composed of the tetrad of catalytic residues. Should be corrected in the manuscript.
- Every part in the manuscript is numbered 1.
- Fig 4 and 5 - font size should be increased in the figures since it is basically impossible to read labels.
Author Response
We kindly thank all the suggestions for the improvement of the manuscript. Please see the attachment.
- Line 122-123: ".... the crystal structure of the 5-LOX/NDGA complex which is composed of tetrad ...". It is not the crystal structure, which is composed of tetrad of catalytic residues. It is the active site, which is composed of the tetrad of catalytic residues. Should be corrected in the manuscript.
Response: We thank the commentary; we made the change in the line to:” …the crystal structure of the 5-LOX/NDGA complex (pdb code: 6N2W) which active site is composed of a tetrad of catalytic residues…”
Summary changes: The text was modified on line 122-123.
- Every part in the manuscript is numbered 1.
Response: We thank the commentary. All the number sections where corrected.
Summary changes: The number 1 on each section was corrected to sequential numbers.
- Fig 4 and 5 - font size should be increased in the figures since it is basically impossible to read labels.
Response: We thank the suggestion. All the font size from the figures 4-5 where increased.
Summary changes: Figures 4 and 5 fonts were changed.